# Management of IgA Nephropathy in Pediatric Patients

**DOI:** 10.3390/children9050653

**Published:** 2022-05-02

**Authors:** Sophie Schroda, Martin Pohl

**Affiliations:** Department of General Pediatrics, Adolescent Medicine and Neonatology, Faculty of Medicine, Medical Center, University of Freiburg, Mathildenstrasse 1, 79106 Freiburg, Germany; sophie.schroda@uniklinik-freiburg.de

**Keywords:** kidney, IgA nephropathy, biopsy, glomerulonephritis, proteinuria

## Abstract

The onset of IgA nephritis in childhood and adolescence often develops into chronic glomerulonephritis with declining renal function. Although these long-term consequences are known, there is still a lack of evidence-based treatment recommendations in this age group. We report data from 22 pediatric patients who were biopsied to confirm the diagnosis of IgAN at our clinical center. 14 of them were treated with corticosteroids according to the recommendations for IgA nephritis vasculitis of the German Society of Pediatric Nephrology (GPN). Improvement was achieved in the majority of all cases, with a significant reduction in proteinuria five months after initiation of therapy. Our data suggest that treatment regimens for acute IgA nephritis and IgA vasculitis nephritis may be unified and are discussed in the context of current studies.

## 1. Introduction

IgA nephritis (IgAN) comprises a spectrum of heterogeneous courses, ranging from rapidly resolving glomerulonephritis to progressive loss of renal function [1]. Onset in childhood and adolescence often leads to transition to chronic nephritis and long-term impaired prognosis [2]. Most patients with IgAN who require renal replacement therapy are young adults [3,4] who possibly developed the disease in childhood.

While mild manifestations of IgAN allow watchful waiting combined with renoprotective therapy, the pediatric nephrologist faces a difficult decision in more aggressive cases, as no evidence-based guidelines have been formulated to date. The Kidney Disease Improving Global Outcomes (KDIGO) guidelines recommend the use of renin-angiotensin system blockade in children with proteinuria > 0.2 g/g creatinine. Despite widespread use of immunosuppressive therapy, especially glucocorticoids, there is no international consensus on their indication in more severe cases as strong trial-based evidence is still missing [5,6].

The aim of this small retrospective study and literature review is to highlight the challenges of pediatric IgAN in clinical practice and to discuss them in the context of current studies.

## 2. Materials and Methods

We retrospectively reviewed the charts of 22 children who were diagnosed with IgAN after renal biopsy in our pediatric nephrology center from 2001 to 2021.

Clinical findings such as hematuria and hypertension were assessed. Microhematuria was defined as microscopic evidence of more than 5 erythrocytes in a high-power field; macrohematuria meant a visible change of urine color. Hypertension was defined as a systolic and/or diastolic blood pressure above the 95th percentile according to age, height, and gender following the KiGGS-study [7]. Proteinuria was estimated using the urine protein to creatinine ratio (UPCR) in g protein/g creatinine. A ratio of >0.2 is considered as proteinuria and >3.5 as nephrotic range proteinuria. For glomerular filtration rate estimation (eGFR), the Schwartz formula using body length and plasma creatinine was used [8].

The biopsy specimens were analyzed by a pathologist according to the Oxford classification (MEST score). For 14 biopsies preceding the updated 2017 classification [9], the C-score was derived from the histopathological description. A minimum of 8 glomeruli per sample was needed for biopsy evaluation.

Follow-ups were carried out at four points in time: after 4 (2–6) weeks, 5 (3–7) months, one year (11–18 months), and two to three years (20–38 months). For patients with an observation period of more than 3.5 years, final follow-up data were also added (mean 5.8 years). Due to missing data points, the size of the patient collective varies as indicated. One patient with end-stage renal disease at initial presentation received kidney transplantation; all posttransplant clinical parameters were excluded from the study.

Treatment modalities included renoprotection with angiotensin converting-enzyme inhibitor (ACEI) or angiotensin II receptor blocker (ARB), immunosuppressive therapy (prednisone, cyclophosphamide, cyclosporine A, tacrolimus, or mycophenolate mofetil), plasmapheresis, or hemodialysis, as well as kidney transplantation. The treatment followed the recommendations of the German Society for Pediatric Nephrology (GPN) for IgAVN [10] and aimed for complete remission as documented by proteinuria < 0.2 g protein/g creatinine and normalized eGFR > 90 mL/min/1.73 m^2^. Glucocorticoid therapy was restricted to 8 weeks and then replaced by alternative immunosuppression where necessary to reduce steroid toxicity. The side effects of steroid therapy were monitored by regular determinations of blood pressure, weight, and glucosuria in urine dipstick.

Results were expressed as means with standard deviation and minimum–maximum values. Parameters such as proteinuria and eGFR over time were compared using the Wilcoxon signed-rank test, parameters of differently treated groups were compared by Mann–Whitney U test, and *p*-values < 0.05 were considered statistically significant. Analyses were performed using SciPy for Python [11]. The study was approved by the Ethics Committee of the University of Freiburg (22-1059).

## 3. Results

### 3.1. Diagnosis

We studied 22 children and adolescents diagnosed with IgAN from 2001 to 2021 in the Department of Pediatrics, Adolescent Medicine and Neonatology, at the University Hospital of Freiburg, Germany. Table 1 outlines the clinical characteristics and the pathological findings at biopsy. Our study group was male-dominated (64%). The mean age at symptom onset was 10.2 ± 3.7 years. The average time to biopsy was 41 weeks (2 days–10 years). At the time of biopsy, the average age was 11.0 ± 3.5 years.

Half of the children and adolescents initially presented with hypertension; 32% showed microhematuria and 68% macrohematuria. The mean proteinuria at biopsy was 3.2 ± 3.7 g/g creatinine and average eGFR 81.3 ± 49.7 mL/min/1.73 m^2^; 32% showed severely impaired renal function with an eGFR < 60 mL/min/1.73 m^2^, and in 41% proteinuria extended into the nephrotic range. All children were followed for a median of 3.5 years (2 months-10 years).

The average number of glomeruli in the biopsy sample was 31.8 ± 24.1 (8–107) per biopsy. The Oxford classification [9] showed 68% of children with mesangial proliferation (M1), 55% with endocapillary proliferation (E1), 64% with segmental sclerosis/adhesion lesion (S1), 5% with moderate tubular atrophy/interstitial fibrosis (T1 25–50% of cortical area involved), none with severe tubular atrophy/interstitial fibrosis (T2 > 50% of cortical area involved), 59% with crescents (C1 < 25% of glomeruli), and none with more than 25% (C2).

### 3.2. Treatment

As described in Table 2, 95% of the 22 children received therapy with ACE inhibitors or AT receptor blockers and another 14 children (64%) underwent i.v. corticosteroid therapy. A total of 6 children (27%) received additional immunosuppressants (mycophenolate mofetil, cyclosporine A, cyclophosphamide, tacrolimus, budesonide). Hemodialysis was performed in 2 children (9%); one of them also required plasmapheresis. One patient (5%) with end-stage renal disease at initial presentation required renal transplantation.

Steroid therapy involved three intravenous administrations of methylprednisolone (or equivalent dose of i.v. prednisone) at a dose of 300 mg/m^2^ body surface area (max. 3 × 500 mg) every 48 h, followed by prednisone p.o. in descending doses (initially 4 weeks with 60 mg/m^2^ BSA/48 h, max. 80 mg, then 4 weeks 40 mg/m^2^ BSA/day, max. 60 mg). This is consistent with the treatment of IgA vasculitis nephritis (IgAVN) patients, as recommended by the German Society of Pediatric Nephrology (GPN) in 2013. In patients with deteriorating eGFR despite intensified immunosuppression (corticosteroid therapy and cyclophosphamide), plasmapheresis is used as rescue therapy [10].

Although treatment decisions were made on a case-by-case basis, the most important factors to initiate a steroid therapy were the evidence of crescents in the biopsy specimen (86% in the treated vs. 13% in non-treated group), nephrotic range proteinuria (57% vs. 13%), or severely impaired renal function (43% vs. 13% with an eGFR < 60 mL/min/1.73 m^2^).

### 3.3. Follow-Up and Outcome

The collected data (proteinuria and eGFR) of five follow up visits are shown in Figure 1. Table 3 indicates the corresponding means, standard deviations, and ranges.

Five months after start of therapy, protein excretion decreased significantly (*p* < 0.001) from 3.2 ± 3.7 (0.29–12.5) to 0.5 ± 0.6 (0.08–2.1) g/g creatinine. However, the improvement in eGFR from 81.3 ± 49.7 (8.9–229.4) to 92.7 ± 32.4 (8.4–162.8) mL/min/1.73 m^2^ was not significant.

Considering only the patients who received i.v. steroid therapy after diagnosis, there was also a significant reduction in proteinuria (*p* < 0.001) from 4.0 ± 3.4 (0.29–12.5) to 0.4 ± 0.5 (0.08–1.7) g/g creatinine after 5 months of follow-up. This subgroup is shown as light blue boxplots in Figure 1.

As shown in Table 4, after 5 months, 25% (5/20) of patients were in complete remission (proteinuria < 0.2 g/g creatinine with normal renal function), 57% (12/21) showed proteinuria > 0.2 g/g creatinine, and 0% (0/21) showed nephrotic range proteinuria. Impaired renal function (eGFR < 90 mL/min/1.73 m^2^) was present in 50% (10/20). At later follow-up time points, patient numbers declined, but the percentages of proteinuria and impaired GFR did not change substantially.

## 4. Discussion

Treatment of newly diagnosed IgAN according to the GPN regimen for IgA vasculitis nephritis achieved an improvement in the majority of cases, with a significant reduction in proteinuria 5 months after initiation of therapy. Nonetheless, complete remission was only achieved in 25% of all cases, and additional immunosuppressants were used in six cases (27%). Interestingly, despite worse baseline conditions, eGFR tended to be better in the corticosteroid-treated patient group than in the untreated group during follow-up. After one year, mean eGFR was even significantly better in the treated group (118.6 vs. 86.5 mL/min/1.73 m^2^, *p* < 0.05 in Mann–Whitney U test). However, this study has limitations such as the small number of patients, the limited follow-up time, and the loss of some patients during this period. Therefore, it is primarily suitable for representing the short-term response to therapy.

In the clinical setting, it is difficult to use immunosuppressive therapy with known side effects and unclear efficacy if the long-term renal damage is not foreseeable, especially because guidelines are based on insufficient evidence. Therefore, based on the existing literature on IgAN in childhood, we would like to discuss why, whom, and how to treat.

### 4.1. Why Treat?

Spontaneous remissions of mild IgAN have been described [12] and led to the early assumption that IgAN in childhood is a benign disease. However, progression of IgAN in children to poor outcome (end points end-stage renal disease (ESRD) or decrease in GFR > 50% after more than 4 years of follow-up) was reached in 7.2% [13] and 12.4% [14] in China, 11% in Finland [15], 18.1% in Sweden [16], and 18.5% in Brazil [17]. The onset of the disease in childhood can therefore lead to a significant reduction in quality of life, as it may require renal replacement therapy at young age [18]. Some studies suggest that achieving remission of proteinuria correlates negatively with progression to chronic kidney disease [19] and that an earlier treatment might be beneficial [20,21]. In small randomized controlled pediatric IgAN studies, immunosuppression reduced proteinuria and the development of glomerulosclerosis [22,23]. In our retrospective analysis, immunosuppressive treatment was given in the more severe cases and led to an improvement and to a comparable outcome, as in the clinically and histologically less severe cases.

### 4.2. Whom to Treat?

In order to identify IgAN patients with poor prognosis already at initial diagnosis, many investigations have been conducted to evaluate corresponding risk factors. Early clinical or pathologic features that bear prognostic significance are summarized in Table 5. While several studies agree on a worse prognosis in older patients [20,24], high proteinuria [13,16,25,26,27] or impaired GFR at the time of diagnosis [16,20,26,27], the pathological results according to the Oxford classification are more controversial in their individual significance. The discordance of the studies can be explained by different inclusion criteria, by the timing of the biopsy (school screening programs in several Asian countries lead to earlier diagnosis and thus milder histopathological changes), by inconsistent outcome criteria, and finally by differences in statistical analysis [28].

Based on the *International IgAN Prediction Tool at biopsy* for adults, a version adapted to children and adolescents was developed in 2020 that promises to accurately predict the risk of a 30% decrease in GFR or ESRD [29]. The criteria considered were age, gender, race, height, weight, proteinuria, serum creatinine, blood pressure, and MEST score according to Oxford classification, as well as the use of RASB or immunosuppressants at or prior to biopsy. Such developments raise hope for a standardized assessment of pediatric patients and thus guideline-based therapy in the future.

### 4.3. How to Treat?

The question of how IgAN should be treated is the most controversial. Renoprotective therapy with ACEI or ARB is the basis of any treatment because of its positive effects, not only on hypertension but also on proteinuria and reduction of GFR decline [30]. Its benefits were demonstrated in a randomized placebo-controlled trial (RCT) in children and young people with IgAN (IgACE [31]). A recent review of previous studies [32] found that the use of ACEI and/or ARB in pediatric patients with IgAN appears to be safe and to reduce proteinuria. Nevertheless, it was pointed out that further RCTs with greater methodological rigor and longer follow-up are needed to confidently demonstrate the efficacy and safety of this therapy in a pediatric population.

However, the use of renin-angiotensin system blocking drugs alone does not always show sufficient response. A recent large study in China provides evidence of the benefit from additional immunosuppressive therapy for children with proteinuria ≥ 1 g/day and initial eGFR of >50 mL/min/1.73 m^2^ [13]. The European VALIGA cohort even suggests that corticosteroids reduce the risk of progression, regardless of initial eGFR and in direct proportion to the extent of proteinuria [33]. Steroids represent the most commonly used form of immunosuppression in IgAN, but the route of steroid administration, dose, and duration of use vary among studies, making comparisons difficult. Steroid pulse therapy, as recommended by the KDIGO guidelines, is discussed as a useful addition for the rapidly progressive glomerulonephritis form of IgAN to achieve faster and more potent efficacy. Furthermore, it may allow a steroid-sparing effect with less cumulative toxicity than sustained oral therapy. Still, the long-term efficacy is uncertain [21].

Other immunosuppressive regimens involve cyclophosphamide [34], azathioprine [22,23], mycophenolate mofetil [35], or tacrolimus [36], but the evidence of their use in children is even more scarce. Lastly, a double-blind, placebo-controlled study comparing vitamin E administration with placebo in children for 1–2 years demonstrated a significant reduction in proteinuria in the vitamin E group, although only mild cases of IgAN were included [37].

In summary, the available published experience suggests that early immunosuppressive therapy might have a beneficial effect on the long-term course of IgAN, at least in severe cases, but sufficiently large studies proving this assumption are outstanding. Our data showing improvement of proteinuria and stabilization of eGFR with an IgAVN treatment protocol and the very similar clinical and histological presentation of acute pediatric IgAN and IgAVN [38,39] may justify the use of the same treatment protocol for both disease entities until more evidence-based treatment protocols are established. Therefore, we suggest unifying treatment protocols for pediatric IgAN and IgAVN, which could also simplify patient recruitment for future prospective studies.

## Figures and Tables

**Figure 1 children-09-00653-f001:**
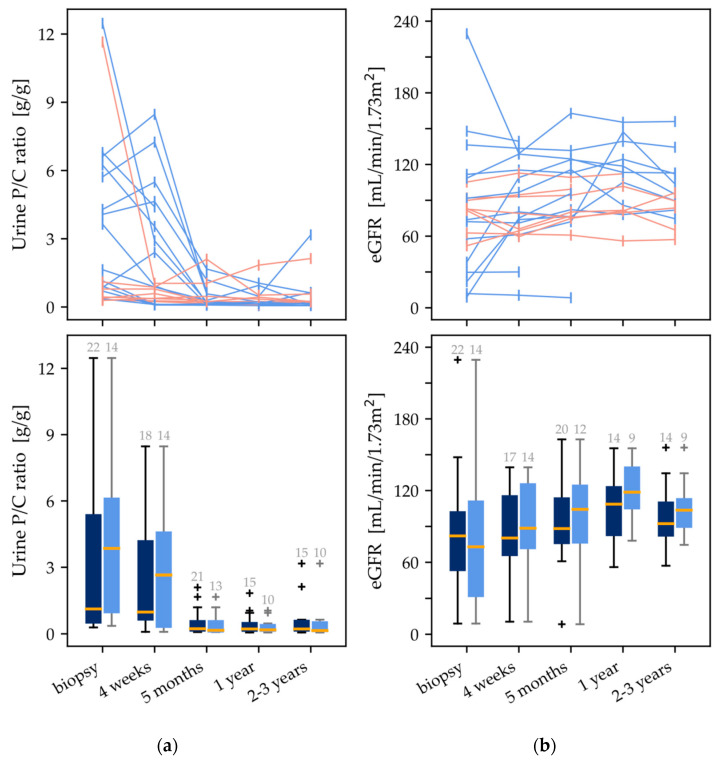
Follow-up data for five points in time: (**a**) urine protein to creatinine ratio (P/C) in g/g; (**b**) eGFR (mL/min/1.73 m^2^). The blue lines represent patients who received i.v. corticosteroid therapy, and the orange lines represent those without. The boxplots below show the data for all patients (dark blue) and for the proportion of patients who received i.v. corticosteroids (light blue). The box-plot outliers are represented by the symbol “+”.

**Table 1 children-09-00653-t001:** Symptom onset and renal biopsy.

Patients	*n* = 22
Gender	14 (64%) males, 8 (36%) females
Age at first symptoms	10.2 ± 3.7 (3.5–15.5)
Age at biopsy	11.0 ± 3.5 (3.5–15.5)
Time to biopsy	41 weeks ± 115 weeks (2 days–10 years)
**Clinical findings**	
Hematuria	7 (32%) micro-, 15 (68%) macrohematuria
Hypertension	11 (50%)
**Laboratory findings**	**(at biopsy)**
Creatinine (mg/dL)	1.5 ± 1.8 (0.18–7.8)
eGFR (mL/min/1.73 m^2^)	81.3 ± 49.7 (8.9–229.4)
eGFR < 60 mL/min/1.73 m^2^	7/22 (32%)
IgA (mg/dL)	234.5 ± 126.2 (98–649)
Urine protein to creatinine ratio	3.2 ± 3.7 g/g (0.29–12.5)
UPCR > 3.5 g/g creatinine	9/22 (41%)
**Kidney biopsy**	
M1	15 (68%)
E1	12 (55%)
S1	14 (64%)
T1	1 (5%)
Crescents (C1)	13 (59%)

**Table 2 children-09-00653-t002:** Treatment.

Treatment	Number (%)
ACE inhibitors/ARB	21 (95%)
Corticosteroid therapy (i.v. + oral)	14 (64%)
Immunosuppressants	6 (27%)
Plasmapheresis and/or hemodialysis	2 (9%)
Transplantation	1 (5%)
**Treatment combinations**	
ACEI/ARB alone	6 (27%)
ACEI/ARB + CS	9 (41%)
ACEI/ARB + CS + IS	4 (18%)
ACEI/ARB + CS + IS + HD	1 (5%)
ACEI/ARB + CS + IS + HD + PP + RT	1 (5%)

ACEI = angiotensin converter enzyme inhibitor, ARB = angiotensin receptor blocker, CS = corticosteroid therapy, IS = immunosuppressants, HD = hemodialysis, PP = plasmapheresis, RT = renal transplant.

**Table 3 children-09-00653-t003:** Follow-up data.

	At Biopsy *n* = 22	4 Weeks after Treatment *n* = 18	After 5 Months *n* = 20	After 1 Year *n* = 14	After 2–3 Years *n* = 14	Last Follow-Up * *n* = 10
Creatinine	1.5 ± 1.8	1.1 ± 1.4	1.0 ± 1.5	0.6 ± 0.2	0.7 ± 0.2	0.9 ± 0.2
(mg/dL)	(0.18–7.8)	(0.32–6.13)	(0.26–7.51)	(0.29–1.04)	(0.3–1.1)	(0.49–1.2)
eGFR	81.3 ± 49.7	87.8 ± 37.1	92.7 ± 32.4	107.1 ± 28.8	96.6 ± 26.4	86.4 ± 19.4
(mL/min/1.73 m^2^)	(8.9–229.4)	(10.5–139.4)	(8.4–162.8)	(56.0–155.3)	(57.1–156.0)	(57.2–130.2)
Proteinuria	3.2 ± 3.7	2.5 ± 2.6	0.5 ± 0.6	0.4 ± 0.5	0.6 ± 0.9	0.4 ± 0.41
(g/g creatinine)	(0.29–12.5)	(0.09–8.47)	(0.08–2.1)	(0.06–1.84)	(0.06–3.17)	(0.08–1.13)

* Patients who were followed for more than 3.5 years, mean value 5.8 years.

**Table 4 children-09-00653-t004:** Outcome.

No. of Patients with…	At Biopsy	After 4 Weeks	After 5 Months	After 1 Year	After 2–3 Years	Last Follow-Up *
Proteinuria > 0.2 g/g	22/22 (100%)	14/18 (78%)	12/21 (57%)	9/15 (60%)	8/15 (53%)	6/10 (60%)
Proteinuria > 3.5 g/g	9/22 (41%)	6/18 (33%)	0/21 (0%)	0/15 (0%)	0/15 (0%)	0/10 (0%)
eGFR < 90 mL/min/1.73 m^2^	13/22 (59%)	9/18 (50%)	10/20 (50%)	5/14 (36%)	7/14 (50%)	5/10 (50%)

* Patients who were followed for more than 3.5 years, mean value 5.8 years.

**Table 5 children-09-00653-t005:** Prognostic factors in pediatric IgAN.

Study	Factors Associated with Poor Prognosis
Yoshikawa et al. 1992 [25] (200 patients, Japan)	Heavy proteinuria at biopsy Diffuse mesangial proliferation, high proportion of glomeruli showing sclerosis, crescents or capsular adhesions, tubulointerstitial changes, subepithelial electron-dense deposits and lysis of the glomerular basement membrane by electron microscopy
Halling et al. 2012 [16] (99 patients, Sweden)	Low GFR, high mean arterial blood pressure and high amount of albuminuria at time of biopsy, low GFR and a high albuminuria during follow-up M1, E1, T1–2, C1
Mizerska et al. 2017 [26] (55 patients, Poland)	Nephrotic-range proteinuria GFR reduction at onset of disease MEST score ≥ 3
Coppo et al. 2017 [24] (261 patients < 23 y, VALIGA European cohort)	Reduction in eGFR of <90 mL/min/1.73 m^2^ at biopsy Proteinuria at >0.4 g/day/1.73 m^2^ M1 in Oxford Classification Older age
Suh et al. 2020 [20] (1154 patients, Korea)	Older age Combined hematuria and proteinuria eGFR < 60 mL/min/1.73 m^2^ Crescents (≥25%)
Wu et al. 2020 [13] (1243 patients, China)	Urinary retinol-binding protein ≥ 0.7 µg/mL Hypertension Hyperuricemia High 24 h protein-excretion Lower initial eGFR High urine C3 levels S1 and T2 lesions

## Data Availability

Not applicable.

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
