# Peer review of "Management of IgA Nephropathy in Pediatric Patients"

_children, 2022, doi:10.3390/children9050653_

Round 1

Reviewer 1 Report

In this article, the authors assessed the clinical parameters of kidney function after treatments in a retrospective cohort of IgA nephropathy children. This article is very clear and well presented. As pointed out by the authors, the cohort is small.

In Table 2, the head of column "(combination possible)" is confusing. It should be "number (%)". The combination of treatments is not described. It would be very informative to see the combination. Does the patient treated with kidney transplantation, previously undergo to dialysis?  What were the clinical criteria for plasmapheresis?

In figure 1, the first graphs show a line for each patient. The authors should indicate the color correspondence for patients treated with immunosuppression.

Author Response

“In this article, the authors assessed the clinical parameters of kidney function after treatments in a retrospective cohort of IgA nephropathy children. This article is very clear and well presented. As pointed out by the authors, the cohort is small.”

Response: Thank you very much for this friendly comment.

“In Table 2, the head of column "(combination possible)" is confusing. It should be "number (%)". The combination of treatments is not described. It would be very informative to see the combination. Does the patient treated with kidney transplantation, previously undergo to dialysis?  What were the clinical criteria for plasmapheresis?”

Response: Thank you for pointing this out. We have revised Table 2 according to your suggestion and replaced “combination possible” by “number (%)”. The five possible treatment combinations have been added in detail under the heading “treatment combinations” in table 2. The patient with kidney transplantation underwent hemodialysis prior to transplantation, as now   shown in table 2. We also performed plasmapheresis in this patient. Plasmapheresis is described in the German Society for Pediatric Nephrology (GPN) treatment recommendation as a rescue therapy performed when eGFR deteriorates under immunosuppressive therapy (corticosteroid therapy and cyclophosphamide). In paragraph 3.2 we added the sentence: “In patients with deteriorating eGFR despite intensified immunosuppression (corticosteroid therapy and cyclophosphamide) plasmapheresis is used as rescue therapy.”

“In figure 1, the first graphs show a line for each patient. The authors should indicate the color correspondence for patients treated with immunosuppression.”

Response: We have implemented this excellent suggestion. In the legend of Figure 1 this sencence was added: “The orange lines represent patients who received i.v. corticosteroid therapy, and the blue lines represent those without.“

Reviewer 2 Report

Schroda et al. presented 22 IgA nephropathy pediatric patients and their treatment. This is a retrospective case series study. I have some suggestions for this manuscript.

  1. What is the treatment goal for IgA nephropathy in pediatric patients? Among IgA patients receiving treatment, 64% of pediatric patients had corticosteroid therapy. Thus, how to evaluate and manage the steroid-associated side effect in pediatric patients?

  1. What is the key message from this retrospective case series study for the reader? In the discussion section, we didn’t see the clinical implication of the current study result.

  1. Is there any plan for treatment protocol for pediatric IgA nephropathy at the University of Freiburg?

Author Response

“1. What is the treatment goal for IgA nephropathy in pediatric patients? Among IgA patients receiving treatment, 64% of pediatric patients had corticosteroid therapy. Thus, how to evaluate and manage the steroid-associated side effect in pediatric patients?”

Response: Thank you very much for this comment. We have added the treatment goal and management of side effects in the Material and Methods section as follows:

“The treatment followed the recommendations of the German Society for Pediatric Nephrology (GPN) for IgAVN [11] and aimed for complete remission as documented by proteinuria < 0.2 g protein / g creatinine and normalized eGFR > 90 ml/min/1.73m². Glucocortiod therapy was restricted to 8 weeks and then replaced by alternative immunsuppression where necessary to reduce steroid toxicity. Side effects of steroid therapy were monitored by regular determinations of blood pressure, weight, and glucosuria in urine dipstick.”

“2. What is the key message from this retrospective case series study for the reader? In the discussion section, we didn’t see the clinical implication of the current study result.”

Response: Thank you for this important question. We have  clarified the clinical implication of the study result, which we formulate in the last paragraph of the manuscript. The case series of 22 pediatric IgAN patients treated with a protocol for acute IgAVN shows improvement of proteinuria and stable eGFR. Therefore, our data and the very similar clinical and histologic presentation of acute pediatric IgAN and IgAVN may justify the use of the same treatment protocol for both disease entities until more evidence-based treatment protocols are established. The small size of our cohort leads us to be cautious with the key message. Nevertheless, we would like to suggest unifying treatment protocols for both disease entities, at least in children.

The following sentence was therefore added to the wording of the key message in the last paragraph of the manuscript: “Therefore, we suggest unifying treatment protocols for pediatric IgAN and IgAVN, which could also simplify patient recruitment in future prospective studies.”

“3. Is there any plan for treatment protocol for pediatric IgA nephropathy at the University of Freiburg?”

Response:  In the treatment of pediatric IgA nephropathy, we follow the recommendations on IgAVN treatment formulated by the German Society for Pediatric Nephrology (GPN).  We added to the material and method section the sentence: “The treatment followed the recommendations of the German Society for Pediatric Nephrology (GPN) for IgAVN [11] and aimed for complete remission as documented by proteinuria < 0.2 g protein / g creatinine and normalized eGFR > 90 ml/min/1.73m².” and “In patients with deteriorating eGFR despite intensified immunosuppression (corticosteroid therapy and cyclophosphamide) plasmapheresis is used as rescue therapy.“

Round 2

Reviewer 2 Report

All comments had been replied well. I have no further suggestions.